# The Association between HDL-C and Subclinical Atherosclerosis Depends on CETP Plasma Concentration: Insights from the IMPROVE Study

**DOI:** 10.3390/biomedicines9030286

**Published:** 2021-03-11

**Authors:** Gualtiero I. Colombo, Vanessa Bianconi, Alice Bonomi, Sara Simonelli, Mauro Amato, Beatrice Frigerio, Alessio Ravani, Cecilia Vitali, Daniela Sansaro, Daniela Coggi, Massimo R. Mannarino, Kai P. Savonen, Sudhir Kurl, Bruna Gigante, Andries J. Smit, Philippe Giral, Elena Tremoli, Laura Calabresi, Fabrizio Veglia, Matteo Pirro, Damiano Baldassarre

**Affiliations:** 1Centro Cardiologico Monzino IRCCS, 20138 Milan, Italy; alice.bonomi@cardiologicomonzino.it (A.B.); mauro.amato@cardiologicomonzino.it (M.A.); beatrice.frigerio@cardiologicomonzino.it (B.F.); alessio.ravani@cardiologicomonzino.it (A.R.); daniela.sansaro@cardiologicomonzino.it (D.S.); elena.tremoli@cardiologicomonzino.it (E.T.); fabrizio.veglia@cardiologicomonzino.it (F.V.); damiano.baldassarre@cardiologicomonzino.it (D.B.); 2Department of Medicine, University of Perugia, 06129 Perugia, Italy; vanessa.bianconi@ospedale.perugia.it (V.B.); massimo.mannarino@unipg.it (M.R.M.); matteo.pirro@unipg.it (M.P.); 3Department of Pharmacological and Biomolecular Sciences, Università degli Studi di Milano, 20133 Milan, Italy; sara.simonelli@unimi.it (S.S.); cevitali@pennmedicine.upenn.edu (C.V.); daniela.coggi@cardiologicomonzino.it (D.C.); laura.calabresi@unimi.it (L.C.); 4Department of Medicine, Division of Translational Medicine and Human Genetics, Perelman School of Medicine, University of Pennsylvania, Philadelphia, PA 19104, USA; 5Foundation for Research in Health Exercise and Nutrition, Kuopio Research Institute of Exercise Medicine, 70100 Kuopio, Finland; kai.savonen@uef.fi; 6Institute of Public Health and Clinical Nutrition, University of Eastern Finland, 70210 Kuopio, Finland; sudhir.kurl@uef.fi; 7Department of Medicine, Karolinska Institutet, 17177 Stockholm, Sweden; bruna.gigante@ki.se; 8Department of Medicine, University Medical Center Groningen, 9700 RB Groningen, The Netherlands; a.j.smit@umcg.nl; 9Department of Endocrinology, Metabolism, and Prevention of Cardiovascular Diseases, Hôpital Pitié-Salpêtrière—Sorbonne University, 75651 Paris, France; philippe.giral@aphp.fr; 10Department of Medical Biotechnology and Translational Medicine, Università degli Studi di Milano, 20129 Milan, Italy

**Keywords:** cholesteryl ester transfer protein, HDL-C, atherosclerosis, single-nucleotide polymorphisms, carotid intima-media thickness

## Abstract

The impact of cholesteryl ester transfer protein (CETP) on atherosclerosis is highly debated. This study aimed to investigate the associations between plasma CETP or CETP genotypes and carotid intima-media thickness (cIMT) and the influence of high-density lipoprotein cholesterol (HDL-C) on these associations. Plasma CETP and HDL-C concentrations were measured in 552 subjects free of any pharmacological treatment from the IMPROVE cohort, which includes 3711 European subjects at high cardiovascular risk. CETP single-nucleotide polymorphisms (SNPs) and cIMT measures (cIMT_max_; cIMT_mean–max_ of bifurcations, common and internal carotids; plaque-free common carotid [PF CC]-IMT_mean_) were available for the full cohort. In drug-free subjects, plasma CETP correlated with HDL-C levels (r = 0.19, *p* < 0.0001), but not with cIMT variables. When stratified according to HDL-C quartiles, CETP positively correlated with cIMT_max_ and cIMT_mean–max_, but not with PF CC-IMT_mean_, in the top HDL-C quartile only. Positive associations between the CETP concentration and cIMT_max_ or cIMT_mean–max_ were found in the top HDL-C quartile, whereas HDL-C levels were negatively correlated with cIMT_max_ and cIMT_mean–max_ when the CETP concentration was below the median (HDL-C × CETP interaction, *p* = 0.001 and *p* = 0.003 for cIMT_max_ and cIMT_mean–max_, respectively). In the full cohort, three CETP SNPs (rs34760410, rs12920974, rs12708968) were positively associated with cIMT_max_. rs12444708 exhibited a significant interaction with HDL-C levels in the prediction of cIMT_max_. In conclusion, a significant interplay was found between plasma CETP and/or CETP genotype and HDL-C in the prediction of carotid plaque thickness, as indexed by cIMT_max_. This suggests that the association of HDL-C with carotid atherosclerosis is CETP-dependent.

## 1. Introduction

Epidemiological studies have consistently shown that a low plasma concentration of high-density lipoprotein cholesterol (HDL-C) is an independent risk factor for atherosclerotic vascular disease [1,2]. Conversely, Mendelian randomization studies did not support a causal link between HDL-C and atherosclerotic cardiovascular events [3]; the attempts to raise HDL-C pharmacologically did not yield the expected outcomes [4]; and there is a U-shaped relationship between HDL-C and cardiovascular disease [5]. This contrasting evidence has generated a great debate on the real protective role of HDL. HDLs were believed to retard the formation of atherosclerotic lesions through several mechanisms [6], the most relevant one being the removal of cholesterol from macrophages within the arterial wall and its transport to the liver for excretion in bile and feces [7]. This so-called reverse cholesterol transport (RCT) may occur through selective hepatocyte uptake of HDL-C by scavenger receptor B1 (SR-B1) (direct pathway), or cholesteryl ester transfer protein (CETP)-mediated transfer of esterified cholesterol, in exchange for triglycerides, from HDLs to apolipoprotein B (apoB)-containing lipoproteins (low-density lipoproteins (LDLs), very low density lipoprotein (VLDLs)), en route to their hepatic clearance (indirect pathway) [8].

Due to its activity in promoting the transfer of cholesteryl esters (CE) and triglycerides between pro- and anti-atherogenic lipoproteins [9], CETP is expected to play a role in the pathophysiology of atherosclerosis. High CETP activity might have pro-atherosclerotic effects, as it lowers circulating HDL-C levels and takes part in the CE enrichment of atherogenic apoB-containing lipoproteins [10]; conversely, it might exert anti-atherosclerotic effects, since the CETP-mediated transport of CE from HDLs to LDLs/VLDLs may favor the hepatic receptor-mediated clearance of atherogenic lipoproteins [11]. To complicate matters, an alternative CETP-mediated CE transfer from LDLs to HDLs has been reported [12].

To date, the role of CETP in atherogenesis remains controversial [13]. High plasma concentrations of CETP have been associated with faster progression of coronary atherosclerosis [14] and increased carotid intima-media thickness (cIMT) [15,16,17]. The association between CETP and atherosclerosis was partially explained by either decreased HDL-C levels or increased plasma triglycerides [18,19]. Other clinical studies, however, did not support a pro-atherosclerotic role of CETP in coronary/carotid artery districts [20,21].

Association studies between CETP genetic variants and atherosclerotic burden have also yielded contrasting results. Three single-nucleotide polymorphisms (SNPs) associated with modestly elevated HDL-C levels showed a weakly inverse association with coronary atherosclerosis [22]. Conversely, a Mendelian randomization analysis linked genetically determined CETP levels with an increased risk of coronary atherosclerosis [23], but another analysis found no association between CETP SNPs and subclinical carotid atherosclerosis [24].

More uncertainties stem from clinical trials using CETP inhibitors [4]. Torcetrapib, dalcetrapib, and evacetrapib, despite producing a significant increase in HDL-C and a decrease in LDL cholesterol (LDL-C), did not significantly affect coronary or carotid artery atherosclerosis progression [25,26] or prevent cardiovascular events [27,28,29]. Anacetrapib, with similar effects on HDL-C and LDL-C, slightly reduced the incidence of major coronary events [30]. Notably, significant regression of coronary atherosclerosis has been reported among torcetrapib-treated patients reaching high levels of HDL-C [31]. This suggests an interaction between CETP and HDL-C in modulating atherosclerosis.

To shed light on the relationship between CETP and atherosclerosis, we explored the association of CETP plasma concentration and/or CETP SNPs with cIMT in a large cohort of subjects at high cardiovascular risk. Further, we specifically investigated the CETP and HDL-C interaction in cIMT modulation.

## 2. Materials and Methods

### 2.1. Participants

All the study participants enrolled were among those recruited in the Carotid Intima-Media Thickness (IMT) and IMT-Progression as Predictors of Vascular Events in a High-Risk European Population (IMPROVE) study, which is a multicenter, prospective cohort study of European individuals at high cardiovascular risk. Design, objectives, methods, eligibility criteria, and baseline evaluation of the IMPROVE study have been previously reported [32]. Briefly, the IMPROVE study involved 7 centers in 5 European countries (Finland, France, Italy, the Netherlands, and Sweden) and recruited a total of 3711 individuals (1095 in Italy, 1050 in Finland, 533 in Sweden, 532 in the Netherlands, and 501 in France). The eligibility criteria were as follows: 54- to 79-year-old men and women who had at least three cardiovascular risk factors (dyslipidemia, hypertension, diabetes, smoking, or family history of cardiovascular disease), who were asymptomatic for cardiovascular disease, and who were free of any conditions that might limit longevity or cIMT visualization [32]. The IMPROVE study was designed following the rules of Good Clinical Practice (GCP) and the ethical principles established in the Declaration of Helsinki. The study protocol (project ID: QLG1-CT-2002-00896, date of funding approval 09 January 2003) was approved by local ethics committees in each study center: i.e., the Institutional Review Board of the Health Department of the Hospital "Ospedale Niguarda Ca’ Granda", Milan, Italy (approval ID: 2042/03, 3 June 2003); the Regional Ethics Review Board at Karolinska Institutet, Stockholm, Sweden (approval ID: Dnr 2003/03-115, 17 February 2003); the Consultative Committee for the Protection of Persons in Biomedical Research (CCPPRB) at Hôpital Pitie Salpétrière, Paris, France (approval ID: CCP61-03, 2003); the Ethics Committee of the Umbrian Health Authorities, Perugia, Italy (approval ID: N 2725/03/A, 06 February 2003); the Medical Ethics Review Committee, Academic Hospital Groningen, Groningen, the Netherlands (approval ID: METc 2003/054, 12 May 2003); the Research Ethics Committee of the University of Kuopio and Kuopio University Hospital, Kuopio, Finland (approval ID: 39/2003, 11 February 2003); the Research Ethics Committee of the University of Kuopio and Kuopio University Hospital, Kuopio, Finland (approval ID: 140/2002, 10 October 2002). Each participant provided two different written informed consents: one for general participation in the study and one for genotyping.

### 2.2. Biochemical Analyses

Blood was collected after an overnight fast. Plasma samples were kept at –80 °C before shipment to Milan and were not thawed before analysis. Serum concentrations of total cholesterol, HDL-C, and triglycerides were measured in a centralized laboratory (Atherosclerosis Research Unit, Karolinska Institute Stockholm, Sweden) by enzymatic methods on an LX Beckman instrument. The Friedewald formula was used to estimate LDL-C concentrations. CETP plasma concentration was assessed in a subgroup of the IMPROVE cohort, consisting of 552 individuals free of any pharmacological treatment. The plasma concentration of CETP was measured by a competitive enzyme-linked immunoassay as previously described [33].

### 2.3. Ultrasonographic Variables

Carotid ultrasonography was performed as previously described [32]. Briefly, the far walls of the left and right common carotids, bifurcations, and internal carotids were visualized in anterior, lateral, and posterior projections and recorded on sVHS videotapes. Assessment of cIMT measures was performed in a core laboratory (Department of Pharmacological and Biomolecular Sciences, University of Milan, Italy) using dedicated software (M’Ath, Metris, France). The highest cIMT value identified among common carotids, bifurcations, and internal carotid arteries in all angles considered (anterior, lateral, and posterior) was defined as the cIMT_max_, while the average of 8 maximal cIMT measures was defined as the cIMT_mean–max_. Of note, these two cIMT variables incorporated atherosclerotic plaques in their measurements. On the other hand, the mean of cIMT measured in plaque-free areas of common carotids (PF CC-IMT_mean_) was considered as a variable reflecting the thickening of the artery wall and a marker of arterial injury, excluding atherosclerotic plaques, as described in [34]. This variable is the average of all plaque-free mean cIMT values obtained from left and right common carotids visualized in their entire length (excluding the 1st cm) with sequential probe movements of 1 cm length, according to the 3 scan angles. The total number of segments visualized ranged from 6 to 24 according to the subject’s length of the neck. The repeatability of cIMT measurements has previously been described [32].

### 2.4. Genotyping and Quality Control

We selected 142 SNPs in the CETP region previously genotyped with the Illumina CardioMetaboChip array in the IMPROVE study [35]. The CardioMetaboChip is a custom genotyping array that interrogates regions identified by meta-analyses of genome-wide association studies (GWAS) of cardiovascular and metabolic traits and diseases. Quality control procedures and individual-level exclusion criteria for the genotype data have been described previously [35]. Multidimensional scaling (MDS) coordinates, reflecting genetic distances between individuals (population substructure), were calculated based on the CardioMetaboChip genotype data. Outliers, identified by inspection of plots of the first three MDS dimensions (MDS1–3), were excluded [35]. Quality control for each SNP and linkage disequilibrium (LD)-based SNP pruning were performed using PLINK v1.07 [36]. Exclusion criteria applied for the CETP SNPs were: genotype call rates <0.95, a departure from Hardy–Weinberg equilibrium (*p* < 0.01), minor allele frequency (MAF) <0.01, and LD coefficient r2 > 0.95. Following these filtering criteria, 3436 individuals and 61 SNPs were included in the association analysis.

### 2.5. Statistical Analysis

Numerical variables were summarized as mean ± standard deviation (SD). Variables with skewed distributions (triglycerides and cIMT measurements) were summarized as median and interquartile range and log-transformed before analyses. Categorical variables were summarized as numbers and percentages. Clinical and anthropometric variables were compared between groups by an unpaired Student’s *t*-test or a chi-square test, as appropriate. Associations between CETP plasma concentration and cIMT measurements, crude or adjusted for potential confounders (age, sex, total cholesterol, HDL-C, log-triglycerides, and latitude), were tested by general linear models (GLMs). Beta values (β) are intended as the change in cIMT measures, expressed in standard deviations, for one SD increase in the CETP concentration. GLMs were also used to evaluate the *p*-values of the interaction terms between CETP plasma concentration and HDL-C levels. In some analyses, HDL-C levels were divided into quartiles according to the distribution in the overall drug-free group at baseline (i.e., ≤39.4 mg/dL [1st quartile], >39.4 and ≤47.5 mg/dL [2nd quartile], >47.5 and <57.5 mg/dL [3rd quartile], and ≥57.5 mg/dL [4th quartile]), whereas in other analyses, HDL-C levels were divided into two groups (i.e., the 4th quartile [high HDL-C levels] and the pooled 1st, 2nd, and 3rd quartiles [normal/low HDL-C levels]). In addition, CETP concentrations were stratified by the median value (i.e., ≥1.38 µg/mL [high CETP concentration] and <1.38 µg/mL [low CETP concentration]). All tests were two-sided, and *p*-values < 0.05 were regarded as significant.

The sample size of 552 subjects yielded 80% statistical power to detect a significant difference in cIMT_max_ between subjects with CETP concentrations above and below the median, greater than 9.5% (corresponding to 0.24 SDs in log scale) in the whole sample, or greater than 15.2% (corresponding to 0.37 SDs in log scale) in each of the two groups with HDL-C above or below 58 mg/dL (top quartile).

Associations of CETP genotype with baseline cIMT variables were investigated by linear regression analyses in the whole IMPROVE study cohort (*n* = 3436), assuming an additive genetic effect (on the log scale) of allele dosage, and adjusting for age, sex, total cholesterol, log-triglycerides, use of lipid-lowering drugs (statins, fibrate, and/or fish oil), and the first three MDS dimensions to account for population stratification. The significance of each SNP–disease association was determined empirically using the max(T) permutation procedure [37,38], which provides strong control of the family-wise error rate correcting for the number of SNPs tested while accounting for the LD between them. Corrected empirical *p*-values (EMP2) < 0.05, as determined using the max(T) permutation procedure (*n* = 10,000 permutations), were considered statistically significant. Furthermore, we tested for the interaction between CETP SNPs and HDL-C strata (4th quartile vs. the pooled 1st, 2nd, and 3rd quartiles) in the prediction of baseline cIMT variables, including all the above covariates in the regression model. Finally, associations between the CETP genotypes and CETP plasma concentration and/or HDL-C levels were investigated in the drug-free subgroup (*n* = 552) only: in this case, nominal *p*-values < 0.0056 were considered statistically significant, accounting for correction for multiple comparisons (Bonferroni). Analyses were performed using the SAS statistical package (SAS Institute Inc., Cary, NC, USA) v9.4. Allele frequency estimation and association tests were performed using PLINK v1.07 [36].

## 3. Results

### 3.1. Characteristics of the Drug-Free Subjects

The clinical characteristics of the 552 (55% men) individuals free of pharmacological treatments, including stratification by the median CETP plasma concentration, are shown in Table 1. Half of the participants were hypertensive, 12% had diabetes, and about 56% were current or former smokers. The range of CETP plasma concentrations was 0.5–2.8 µg/mL. Subjects with CETP concentrations above the median had significantly higher HDL-C levels and lower triglycerides than those who had CETP concentrations below the median. CETP concentration correlated with HDL-C levels (r = 0.19, *p* < 0.0001).

### 3.2. Association between CETP Plasma Concentration and cIMT Variables

In the entire sample of 552 subjects, CETP plasma concentration was not associated with either the carotid ultrasonographic variables that include plaques (cIMT_max_ and cIMT_mean–max_) or with those measured in plaque-free areas (PF CC-IMT_mean_), either in the crude analysis or in the analysis adjusted for age, sex, HDL-C, log-triglycerides, total cholesterol, and latitude (Appendix A). Despite this, significant interactions (CETP × HDL-C quartiles) were found in the associations of CETP plasma concentration with cIMT_max_ or cIMT_mean–max_ (*p*_interaction_ < 0.05 for both; Appendix A) and not PF CC-IMT_mean_ (Appendix A). The effect of CETP on cIMT_max_ and cIMT_mean–max_ appeared to be limited to the upper quartile of HDL-C concentrations (see the trendlines in Appendix A).

The analysis was repeated by stratifying drug-free subjects according to the CETP concentration (above or below the median, i.e., 1.38 µg/mL) and to the HDL-C levels (≥58 mg/dL, top quartile, or <58 mg/dL, pooled quartiles 1–3; Appendix A), adjusting for age, sex, log-triglycerides, total cholesterol, and latitude. An increase in the CETP plasma concentration was associated with an increased cIMT (i.e., CETP exerts a pro-atherosclerotic effect) in subjects with very high HDL-C (upper quartile; subjects with CETP above the median vs. subjects below the median, *p* = 0.014 and *p* = 0.004 for cIMT_max_ and cIMT_mean–max_, respectively; Figure 1A,B). In turn, high levels of HDL-C were associated with low carotid IMT (i.e., HDL-C plays a protective role) only when the CETP plasma concentration was below the median (HDL-C top quartile vs. quartiles 1–3, *p* = 0.016 and *p* = 0.002 for cIMT_max_ and cIMT_mean–max_, respectively; Figure 1A,B). Conversely, subjects with low/normal HDL-C (quartiles 1–3) showed similarly high carotid IMT irrespective of CETP concentrations (subjects with CETP above the median vs. subjects below the median, *p* = 0.077 and *p* = 0.33 for cIMT_max_ and cIMT_mean–max_, respectively). Indeed, HDL-C × CETP interactions were significant (*p*_interaction_ = 0.001 and 0.003 for cIMT_max_ and cIMT_mean–max_, respectively). The effect of HDL-C on PF CC-IMT_mean_ was similar in both CETP subgroups (CETP × HDL-C *p*_interaction_ = 0.87; Figure 1C), i.e., high levels of HDL-C had protective effects regardless of the CETP concentration (HDL-C top quartile vs. quartiles 1–3, *p* = 0.03 and *p* = 0.009 for subjects with CETP below and above the median, respectively). No interaction was found between CETP and total cholesterol, LDL-C, or triglycerides in determining the ultrasonographic variables considered (not shown).

### 3.3. Association between CETP SNPs and cIMT Variables

The full IMPROVE cohort was used to test for associations between CETP genotypes and ultrasonographic measures. In the multivariable analysis (Table 2), adjusting for age, sex, total cholesterol, log-triglycerides, use of lipid-lowering drugs (statins, fibrate, and/or fish oil), and the first three multidimensional scaling coordinates (population genetic substructure), seven SNPs showed nominally significant associations with cIMT_max_, but only three remained significant after correction for multiple testing (rs34760410, rs12920974, and rs12708968).

The T allele of the peak SNP (rs12920974) was significantly associated with cIMT_max_ both in the whole cohort and in the group of subjects with normal/low HDL-C (pooled quartiles 1–3; Figure 2A,B), but not in the group with high HDL-C levels (top quartile). Conversely, rs12444708 and rs9938160 showed nominally significant associations with cIMT_max_ in the top HDL-C quartile and not in the normal/low HDL-C group or the whole cohort, but only rs12444708 remained significant after correction for multiple testing (Figure 2C).

Importantly, there was a significant interaction between CETP rs12444708 genotypes and HDL-C strata (high vs. normal/low) in predicting cIMT_max_ (*p*_interaction_ = 0.0011; Figure 3). Similar results were obtained for cIMT_mean–max_, but not for PF CC-IMT_mean_ (data not shown).

CETP and HDL-C concentrations stratified according to the SNP allele dosage are reported in Appendix A. While no SNP was associated with plasma CETP, three of the abovementioned SNPs (rs72786786, rs173539, and rs3764261) were significantly associated with HDL-C levels.

## 4. Discussion

This study investigated the relationship between either CETP concentration or CETP SNPs and different measures of cIMT in a European cohort. Three main results emerged: (1) there was a significant interaction between CETP concentration and HDL-C levels in predicting cIMT_max_; (2) three CETP SNPs were independent predictors of cIMT_max_ in the whole cohort; (3) one CETP SNP showed a significant interaction with HDL-C plasma levels in cIMT_max_ prediction.

An unprecedented finding of the present study is the significant interaction between CETP and HDL-C levels in the prediction of cIMT_max_ and cIMT_mean–max_. Although CETP levels did not correlate with any of the ultrasonography measures in the entire drug-free group, high CETP concentrations were associated with increased cIMT_max_ and cIMT_mean–max_ in the HDL-C upper quartile subgroup of patients. Conversely, high HDL-C levels were inversely associated with cIMT_max_ and cIMT_mean–max_ in subjects with CETP levels below the median value. Finally, lower levels of HDL-C were associated with high cIMT_max_ and cIMT_mean–max_ independently of CETP plasma concentrations. These results suggest that the impact of CETP on carotid atherosclerosis is not modulated by HDL-C levels. Conversely, the HDL-C impact on carotid atherosclerosis could be influenced by the CETP concentration. Indeed, high CETP levels might frustrate the putative atheroprotective effect of elevated HDL-C. In contrast, the putative pro-atherogenic effect of low HDL-C levels does not seem to be influenced or counterbalanced by CETP concentrations.

From a molecular perspective, some mechanistic speculations may emerge from our findings. One hypothesis is that the coexistence of high CETP and high HDL-C levels might impair CE transport through the direct RCT pathway and lead to a preferential CETP-mediated CE enrichment of apoB-containing lipoproteins, resulting in enhanced atherosclerosis formation. The evidence of an inverse association between CETP and SR-B1 activity [39], along with a direct association between SR-B1 deficiency and increased susceptibility to atherosclerosis, supports this notion [40]. An additional (but not alternative) hypothesis is that high CETP concentrations may promote atherogenesis by impairing other HDL beneficial functions (e.g., antioxidant and anti-inflammatory) [41,42,43].

From a clinical perspective, the evidence of an interaction between CETP and HDL-C plasma concentrations in predicting cIMT might contribute to explaining the disappointing findings on cardiovascular endpoints of randomized clinical trials with CETP inhibitors. These drugs were shown to be extraordinarily effective in reducing CETP activity and increasing HDL-C levels but were also associated with a significant increase in the CETP concentration [4,26,27,29]. Consistently, we found that, among subjects with high HDL-C, high CETP concentrations were associated with a burden of carotid atherosclerosis at least as high as that of subjects with low/medium HDL-C. Conversely, in subjects with high HDL-C levels, low CETP concentrations were associated with the lowest cIMT. Since HDL-C is inversely associated with cIMT in this and other cohorts [32,44], we could assume that the increase in the CETP concentration during pharmacological inhibition of CETP activity might outweigh the potential anti-atherogenic effect of the parallel increase in HDL-C plasma levels. Such an assumption is partly undermined by a post hoc analysis of the ILLUSTRATE trial showing that a significant regression of coronary atherosclerosis occurred among torcetrapib-treated patients with high HDL-C levels [31]. Nonetheless, given the power (80%) and the sample size (*n* = 552) of our study, we cannot fully settle the controversy in the existing literature [25,26,27,28,29,30,31] and support the potential effectiveness of a complete CETP inhibition (i.e., mass and activity) as an anti-atherogenic strategy. Further studies are needed to answer whether an increase in plasma HDL-C levels following a reduction in CETP mass and activity could prevent atherosclerosis progression and related cardiovascular events.

Noteworthy, we did not observe any interaction between CETP and HDL-C in the prediction of PF CC-IMT_mean_, a measure of arterial wall thickening at the common carotid segment. This suggests that the possible harmful interaction between CETP and HDL-C plasma levels does not to apply to all carotid segments. Different susceptibilities of different tracts of arteries to aging and/or injury may be plausible when considering that different vascular beds exhibit distinct biological phenotypes, which may impact the development of atherosclerosis [45].

In multivariable-adjusted analysis, seven CETP SNPs (see Table 2) showed nominal associations with cIMT_max_ in the full IMPROVE cohort. Three of them (rs34760410, rs12920974, and rs12708968) showed positive associations withstanding correction for multiple testing using permutations. To our knowledge, this is the first study to find an association between these three SNPs and atherosclerosis at any arterial district. Whether this association is influenced by HDL-C plasma levels remains uncertain. The lead SNP rs12920974 was positively associated with cIMT_max_ in the full cohort and low-HDL-C group. Two previous studies have shown a significant negative association between rs12920974 and HDL-C levels [46,47]. On the contrary, we did not find a significant association between rs12920974 genotypes and HDL-C levels, or any interaction between rs12920974 and HDL-C-strata in the prediction of cIMT_max_. Hence, rs12920974 appears related to cIMT_max_, but the impact of reduced levels of HDL-C in mediating its pro-atherosclerotic effect remains unclear. Similarly, no interaction was detected between rs34760410 and rs12708968 and HDL-C in the prediction of cIMT_max_.

Three of the seven CETP SNPs (rs72786786, rs173539, and rs3764261) were negatively associated with cIMT_max_ in the IMPROVE cohort, albeit only with a nominal level of significance. Interestingly, the minor allele dosage of each of these three SNPs showed a positive correlation with HDL-C plasma levels in our drug-free group. Consistently, significant positive associations have been found between rs173539, rs3764261, and rs72786786 and HDL-C levels in three large meta-analyses/GWAS, which included community-based cohorts [48], individuals of European descent [49], and healthy, long-lived subjects [50], respectively.

An interesting novel finding, stemming from the stratification of the full IMPROVE cohort according to HDL-C levels, was the detection of a CETP SNP (rs12444708) positively associated with cIMT_max_ in the HDL-C upper quartile subgroup only, withstanding correction for multiple testing. In addition, we observed a significant interaction between this CETP SNP and HDL-C-strata in predicting cIMT_max_. Indeed, subjects with the rs12444708-TT genotype had significantly higher cIMT_max_ than those with the rs12444708-CC genotype only in the HDL-C top quartile. Given that sleep deprivation contributes to cardiovascular disease, a recent genome-wide, multi-ancestry meta-analysis investigated the influence of habitual sleep duration on genetic associations with blood lipid traits and found that rs12444708 was among the lead SNPs positively associated with HDL-C, showing a significant interaction effect with sleep duration on HDL-C levels [51]. Accordingly, a relationship between CETP rs12444708 and carotid atherosclerosis influenced by HDL-C may be inferred.

An additional finding is that none of the aforementioned SNPs were significantly correlated with the CETP plasma concentration. Although comparisons of CETP concentrations between carriers of these polymorphisms suffered from a limited sample size due to the low minor allele frequency, this result suggests that the reported association between three CETP SNPs (rs34760410, rs12920974, and rs12708968) and cIMT_max_ is independent of the CETP concentration. Whether these SNPs are unable to modulate the CETP concentration, or whether unmeasured confounding factors interfere with the plasma concentration of CETP, affecting its association with these SNPs, remains uncertain.

The present study has some limitations. First, we did not test for CETP activity, which is a more reliable measure of CE transport than the CETP concentration. Nonetheless, a strong positive correlation between the CETP concentration and CETP activity has been previously reported. Second, we did not test for all CETP gene variants, which might have compromised the possibility to better evaluate the relationship between CETP and atherosclerosis. Third, a residual confounding effect due to additional variables (e.g., socio-environmental and behavioral factors) beyond those included in the multivariable analyses cannot be excluded. Finally, the inclusion criteria might limit the generalizability of the study results to the general population.

## 5. Conclusions

The present study suggests the existence of a relationship between CETP and carotid atherosclerosis, as supported by the independent association of three CETP SNPs (rs34760410, rs12920974, rs12708968) with cIMT_max_. The relation of one additional CETP SNP (rs12444708) with carotid atherosclerosis appears to be HDL-C-dependent, in that it was associated with a significant pro-atherosclerotic effect only in the presence of high HDL-C levels. A possible pro-atherogenic role of the combination of high CETP/high HDL-C plasma concentrations—i.e., an impairment of HDL-C protective actions—is supported by the interaction between CETP and HDL-C in predicting cIMT_max_. According to the literature, the reduction in CETP activity and the increase in HDL-C levels induced by using CETP inhibitors [26,27,29] are paralleled by a significant increase in the CETP concentration [26]. Such an increased CETP concentration may have a potential pro-atherogenic effect outweighing the atheroprotective function of high HDL-C levels. Therefore, reducing both the CETP concentration and activity could be a promising anti-atherosclerotic strategy, as has been reported for CETP antisense oligonucleotide inhibitors [52].

## Figures and Tables

**Figure 1 biomedicines-09-00286-f001:**
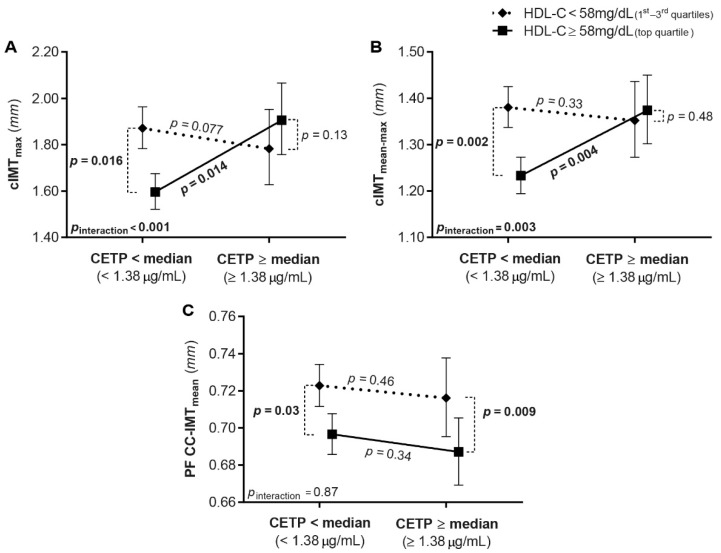
CETP × HDL-C interaction in predicting cIMT. Drug-free subjects (*n* = 552) were stratified according to high-density lipoprotein cholesterol (HDL-C; upper vs. pooled quartiles 1–3) and cholesteryl ester transfer protein (CETP; above vs. below the median value) plasma concentrations. CETP was positively associated with (**A**) maximal carotid intima-media thickness (cIMT_max_) and (**B**) the average of maximal cIMT measurements (cIMT_mean–max_), in the top HDL-C quartile only. HDL-C was negatively correlated with cIMT_max_ and cIMT_mean–max_ when CETP concentration was below the median. There was a significant CETP × HDL-C interaction. (**C**) HDL-C was negatively correlated with the average plaque-free common carotid IMT (PF CC-IMT_mean_) irrespective of CETP concentration (i.e., no interaction between CETP and HDL-C). The analyses were adjusted for age, sex, latitude, total cholesterol, and log-triglycerides.

**Figure 2 biomedicines-09-00286-f002:**
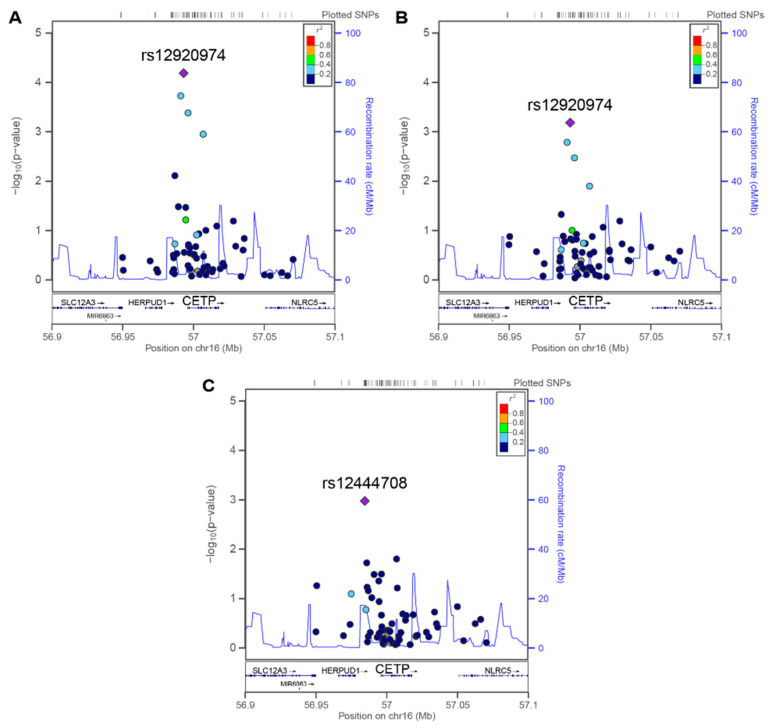
Regional plots of SNPs in the CETP region with evidence of an association with cIMT_max_. Each circle represents a single-nucleotide polymorphism (SNP) plotted with its association *p*-value (on a −log10 scale) as a function of genomic position. (**A**) Plot from the entire IMPROVE cohort. Plots for (**B**) the pooled 1st, 2nd, and 3rd quartiles and (**C**) the 4th quartile of HDL-C plasma concentration. For each association analysis, the lead SNP is represented by a purple diamond. The color of all other SNPs indicates the degree of linkage disequilibrium with the lead SNP (see r2 values on the right). Analyses were adjusted for age, sex, cholesterol, log-triglycerides, use of fibrate and/or fish oil and/or statins, and MDS1–3 (population substructure). Abbreviations are the same as those in Figure 1.

**Figure 3 biomedicines-09-00286-f003:**
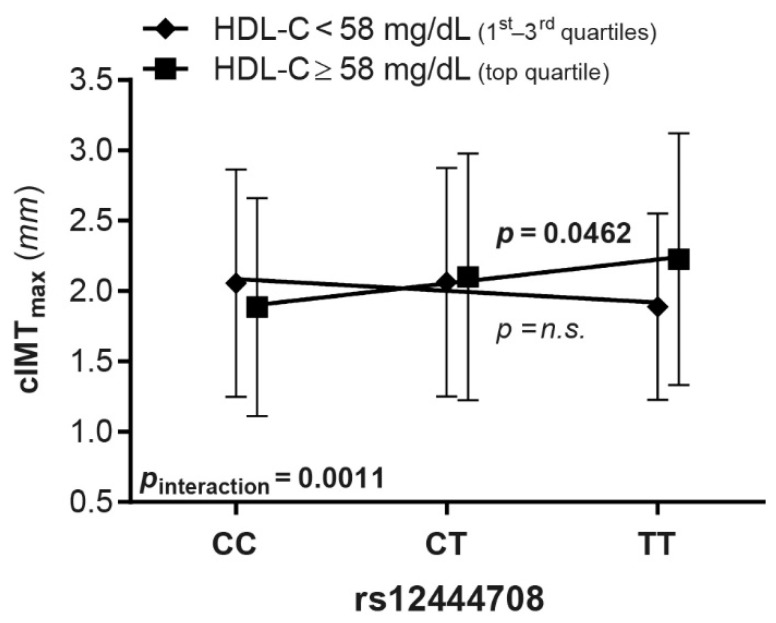
Interaction between the CETP rs12444708 genotype and HDL-C plasma levels in the prediction of cIMT_max_. This SNP was associated with cIMT_max_ only in patients belonging to the top quartile of HDL-C. The analysis was adjusted as in Figure 2. Abbreviations are the same as those in Figure 1.

**Table 1 biomedicines-09-00286-t001:** Characteristics of individuals, free of any pharmacological treatment, stratified by CETP plasma concentrations (above or below the median).

	Whole Sample (*n* = 552)	CETP < 1.38 µg/mL (*n* = 274)	CETP ≥ 1.38 µg/mL (*n* = 278)	*p*-Value
Age, years	63.8 ± 5.0	64.0 ± 5.0	63.6 ± 4.9	0.43
Male sex, *n* (%)	304 (55.1)	151 (55.1)	153 (55.0)	0.53
Hypertensive, *n* (%)	271 (49.1)	139 (50.7)	132 (47.5)	0.25
Diabetes, *n* (%)	67 (12.1)	30 (10.9)	37 (13.3)	0.24
Smoking status				
Never, *n* (%)	240 (43.5)	128 (46.7)	112 (40.3)	
Former, *n* (%)	195 (35.3)	97 (35.4)	98 (35.3)	0.13
Current, *n* (%)	117 (21.2)	49 (17.9)	68 (24.5)	
BMI, kg/m^2^	26.3 ± 3.8	26.2 ± 3.5	26.4 ± 4.1	0.59
Total cholesterol, mg/dL	233 ± 39	231 ± 39	234 ± 39	0.49
LDL cholesterol, mg/dL	158 ± 34	156 ± 35	159 ± 33	0.37
HDL cholesterol, mg/dL	50 ± 15	48 ± 15	52 ± 15	**0.007**
Triglycerides, mg/dL^a^	110 (77–158)	119 (82–170)	102 (74–142)	**<0.0001**
CETP, µg/mL	1.40 ± 0.33	1.14 ± 0.18	1.65 ± 0.23	-
cIMT_max_	1.76 (1.35–2.31)	1.76 (1.39–2.31)	1.84 (1.35–2.31)	0.87
cIMT_mean–max_	1.32 (1.12–1.57)	1.31 (1.13–1.54)	1.33 (1.12–1.64)	0.69
PF CC-IMT_mean_	0.72 (0.65–0.78)	0.72 (0.65–0.78)	0.72 (0.65–0.76)	0.26

CETP, cholesteryl ester transfer protein; LDL, low-density lipoprotein; HDL, high-density lipoprotein; cIMT_max_, highest carotid intima-media thickness (IMT) value identified among common carotids, bifurcations, and internal carotid arteries; cIMT_mean–max_, average of 8 maximal carotid IMT measurements; PF CC-IMT_mean_, mean of IMT measured in plaque-free areas of common carotids. Quantitative variables are expressed as mean ± standard deviation or median and interquartile range (Q1–Q3), categorical variables as counts (*n*) and percentage (%). Between-group comparisons were performed by a Student’s *t*-test or chi-square test, as appropriate. ^a^ Median and interquartile range calculated after log-transformation. Significant *p*-value are highlighted in bold.

**Table 2 biomedicines-09-00286-t002:** Associations between single-nucleotide polymorphisms (SNPs) at CETP locus and cIMT_max_ in the full IMPROVE cohort and in subjects stratified by HDL-C quartiles (pooled quartiles 1–3 vs. top quartile).

			Full Cohort (*n* = 3436)			HDL-C < 58 mg/dL (*n* = 2673)			HDL-C ≥ 58 mg/dL (*n* = 763)			Interaction	
SNP	Position	A1	β	*p* _uncorr_	EMP2	β	*p* _uncorr_	EMP2	β	*p* _uncorr_	EMP2	β	*p*-Value
rs12444708	56950570, intergenic	T	0.005	0.3410	1	−0.005	0.4592	1	0.037	**0.0011**	**0.0462**	0.041	**0.0011**
rs9938160	56950678, intergenic	C	−0.002	0.6733	1	0.004	0.3746	1	−0.021	**0.0205**	0.5504	−0.025	**0.0112**
rs72786786	56951602, intergenic	A	−0.011	**0.0085**	0.3084	−0.009	0.0515	0.8517	−0.015	0.0758	0.9403	−0.005	0.6148
rs173539	56954132, intergenic	T	−0.009	**0.0360**	0.7479	−0.006	0.1826	0.9993	−0.014	0.1043	0.9788	−0.007	0.4636
rs34760410	56955723, intergenic	T	0.023	**0.0002**	**0.0100**	0.022	**0.0018**	0.0740	0.028	**0.0358**	0.7430	0.004	0.7794
rs12920974	56959113, promoter	T	0.016	**0.0001**	**0.0034**	0.016	**0.0007**	**0.0310**	0.019	**0.0479**	0.8311	0.001	0.9171
rs3764261	56959412, promoter	A	−0.009	**0.0372**	0.7587	−0.007	0.1691	0.9980	−0.013	0.1254	0.9892	−0.006	0.5493
rs12708968	56960907, promoter	C	0.021	**0.0005**	**0.0213**	0.019	**0.0037**	0.1458	0.026	**0.0351**	0.7363	0.006	0.6664
rs12708974	56971638, intron 7	T	0.019	**0.0012**	0.0553	0.016	**0.0137**	0.4151	0.030	**0.0171**	0.4945	0.011	0.4152

SNP, single-nucleotide polymorphism; CETP, cholesteryl ester transfer protein; cIMT_max_, highest carotid IMT value identified among common carotids, bifurcations, and internal carotid arteries; HDL-C, high-density lipoprotein cholesterol. Position indicates genomic position based on NCBI genome Build 38.p12; A1, minor allele; p_uncorr_, uncorrected asymptotic *p*-value; EMP2, corrected empirical *p*-value after 10,000 permutation tests. Associations were adjusted for age, sex, cholesterol, log-triglycerides, use of fibrates and/or fish oil and/or statins, and MDS1–3 (population substructure). Significant *p*-value are highlighted in bold.

## Data Availability

The data presented in this study are available on request from the corresponding author. The data are not publicly available due to ethical reasons.

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
