# Peer review of "The Association between HDL-C and Subclinical Atherosclerosis Depends on CETP Plasma Concentration: Insights from the IMPROVE Study"

_biomedicines, 2021, doi:10.3390/biomedicines9030286_

Round 1
Reviewer 1 Report
In the submitted study, Colombo and colleagues analyse the impact of CETP concentration and genetic variations on atherosclerosis using IMT as a proxy for this disease. While determining concentration instead of activity is a certain drawback of the study, it is fully offset by the impressive size of the cohort, which permitted several novel valid and interesting observations. I have no major objections with regard to the study design and applied methodology. While the manuscript is generally well-written, I would suggest condensing the discussion. Basing on the known CETP structure could the authors speculate, what could be the impact of CETP polymorphisms on its enzymatic activity in context of HDL metabolism and IMT?
Author Response
We are thankful to the Reviewer for the positive comment.
According to the Reviewer’s suggestion the discussion has been condensed (30 lines have been eliminated) and focused on the more relevant results.
In our study three CETP SNPs (rs34760410, rs12920974, and rs12708968) showed a positive association with cIMTmax, withstooding correction for multiple testing using permutations. To our knowledge, the relationship between these three CETP polymorphisms and CETP activity has never been directly investigated. Besides, it was not predictable on the basis of our results, as none of the aforementioned SNPs was associated with HDL-C (high CETP activity correlates with reduced HDL-C levels) nor with CETP mass (a positive correlation between CETP concentration and CETP activity has been previously reported). However, based on literature data a negative association exists between one of them (rs12920974), which is located in the CETP gene promoter region, and HDL-C. Therefore, a positive association between this CETP SNP and CETP activity, possibly explaining its positive association with carotid atherosclerosis, might be inferred.
Reviewer 2 Report
The authors analyzed the cohort data for cross-sectional analysis for the factors on carotid intima/media thickness, IMPROVE Study, focusing on HDL-C and CETP. They found marginal relationship between carotid wall thickness and CETP via HDL-C. The data may support the hypothesis that plasma CETP activity is a factor to decrease HDL-C so that low CETP activity is rather beneficial with respect to atherosclerosis risk. Since the power is very marginal, the results may not directly lead to full support of the relevance of developing CETP inhibitors as antiatherogenic reagents. The data themselves are descriptive but solid. However, the discussion in the manuscript does not fully cover the controversy about justification of inhibiting CETP.
Author Response
We thank the Reviewer for the constructive comment.
As pointed out by the Reviewer, at least two factors may dampen the hypothesis that full CETP inhibition (i.e., mass and activity) may be an anti-atherosclerotic strategy: 1) in this study the association between carotid atherosclerosis and CETP via HDL-C had a marginal power and 2) significant controversy emerges in existing literature regarding the impact of CETP inhibition on atherosclerosis. Accordingly, the discussion has been revised by adding the following section: “Nonetheless, given the marginal power of our study results and controversy in the existing literature [25-31], we cannot fully support the relevance of full CETP inhibition (i.e., mass and activity) as an anti-atherogenic strategy. Additional studies are awaited responding as to whether an increase of plasma HDL-C levels following reduction of both CETP mass and activity might prevent atherosclerosis appearance and progression as well as clinical atherosclerosis-related end-points.”